# Do Breastfeeding History and Diet Quality Predict Inhibitory Control at Preschool Age?

**DOI:** 10.3390/nu13082752

**Published:** 2021-08-10

**Authors:** Yvonne Willemsen, Roseriet Beijers, Alejandro Arias Vasquez, Carolina de Weerth

**Affiliations:** 1Department of Cognitive Neuroscience, Donders Institute for Brain, Cognition and Behaviour, Radboud University Medical Center, 6525 EN Nijmegen, The Netherlands; roseriet.beijers@ru.nl (R.B.); Carolina.deWeerth@radboudumc.nl (C.d.W.); 2Behavioural Science Institute, Radboud University, 6525 GD Nijmegen, The Netherlands; 3Donders Center for Medical Neuroscience, Department of Psychiatry and Human Genetics, Radboud University Medical Center, 6525 GA Nijmegen, The Netherlands; Alejandro.AriasVasquez@radboudumc.nl

**Keywords:** breastfeeding, diet, inhibition, executive function, preschool

## Abstract

Inhibitory control is the ability to control impulsive behavior. It is associated with a range of mental and physical health outcomes, including attention deficit hyperactivity disorder and substance dependence. Breastfeeding and healthy dietary patterns have been associated with better executive functions, of which inhibitory control is part. Additionally, breastfeeding has been associated with healthy dietary patterns. Following our preregistration in the Open Science Framework, we investigated the associations between breastfeeding history and inhibitory control at preschool age, with habitual diet quality as a potential mediating factor. A total of 72 families from a longitudinal study participated at child age 3. Breastfeeding questionnaires were administered at 2, 6, and 12 weeks, and at 12 and 36 months. Six inhibitory control tasks were performed during a home visit, and questionnaires were filled in by both parents. Diet quality at age 3 was assessed via three unannounced 24-h recalls. Structural equation modelling was performed in R. This study did not provide evidence that breastfeeding history is associated with inhibitory control in 3-year-old children. Furthermore, diet quality at age 3 did not mediate the link between breastfeeding history and inhibitory control. Previous studies have investigated broader aspects of inhibitory control, such as executive functions, and used different methods to assess nutritional intake, which might explain our differential findings. Our findings contribute to the growing literature on associations between nutrition and behavior. Future replications with larger and more diverse preschool samples are recommended.

## 1. Introduction

Inhibitory control is the ability to control impulsive behavior [1], and is an integral part of higher-order executive functioning (EF) [2]. Inhibitory control has been shown to predict a range of health, wealth and public safety outcomes including physical health, substance dependence, personal finances, and criminal offending outcomes [3]. Furthermore, studies have associated poor inhibitory control with psychopathology, such as attention deficit hyperactivity disorder (ADHD), and child internalizing and externalizing problems [4,5,6]. Several factors have been associated with higher levels of inhibitory control, including high socioeconomic status, high parenting quality, and genetic factors [7,8,9]. Nutrition has also been associated with inhibitory control, though its exact role remains unclear [10]. One biological pathway hypothesized to be involved in the association between breastfeeding, diet and inhibitory control is the microbiota-gut-brain axis, which is a bidirectional communication route between bacteria in the gut and the brain [11]. Gut microbiota are in constant interplay with diet; thus it is possible that the gut microbiota moderate the effects of diet on the brain and potentially affect behavior [11,12]. As inhibitory control develops quickly early in life, early life nutrition might be especially important. This paper investigated the associations between breastfeeding history, habitual diet, and preschoolers’ inhibitory control.

After birth, breast milk is often the first source of nutrition for infants. A continuously increasing body of research shows that early life nutrition shapes the brain, hence affecting its function throughout development, and having major lifelong effects on cognition and behavior [13]. Thus, breastfeeding plays an important role in health and early development [14,15]. For example, Bar et al. (2016) reviewed studies investigating benefits of breastfeeding in relation to cognitive development, and attention-deficit/hyperactivity disorder (ADHD) in preschool and school-age children. They found that children who were breastfed longer than 6 months had better cognitive outcomes compared to children who were breastfed for shorter times [16]. Furthermore, a meta-analysis concluded that breastfeeding may reduce the risk of ADHD in 3- to 17-year-old children [17]. However, while one prospective longitudinal study including 500 preschoolers found that EF (assessed by validated questionnaires) was better when they were breastfed longer [18], another prospective longitudinal study including 180 children found no relation between breastfeeding and EF (assessed with cognitive tests) in 6- to 7-year-old children [19]. Additionally, a recent cross sectional study by Lopez et al. (2021) also found no evidence for an association between breastfeeding and executive functioning (assessed with cognitive tests) in 9,116 9-10-year-old children [20]. To our knowledge, no study has specifically examined associations between breastfeeding history and inhibitory control in 3-year-olds.

The association between habitual diet and child EF, including inhibitory control, is equivocal (see review Egbert et al., 2019) [10]. For example, the two cross-sectional studies by Levitan et al. (2015) (*n* = 193) and Pieper and Laugero (2013) (*n* = 29) performed in preschool-aged children, who assessed spontaneous food choices in an experimental setting, found no relation between inhibitory control and calories or protein consumed [21,22]. Note however, that Levitan et al. (2015) found that higher intake of sugars was associated with worse inhibitory control in preschoolers. Additionally, a review by Cohen and colleagues (2016) including 21 studies, showed positive associations between overall higher diet quality and executive functioning in studies examining the effects of long term diet on executive functioning in children aged 5–17 years [23]. Intake of fruits, vegetables, fish, and whole grain products were associated with improved executive functions, while higher intake of snacks and sugar-sweetened beverages were associated with worse executive functions [23]. As the development of problem behavior typically starts around preschool age, and behavior problems at this early stage increase the risk for poor developmental outcomes [5], it is important to investigate the early determinants of preschoolers’ inhibitory control.

While breastfeeding might play a role in the development of child inhibitory control, its role has only been investigated in relation to ADHD and EF, to our knowledge. Moreover, the role of habitual diet predicting preschoolers’ inhibitory control received little attention. Since breastfeeding is predictive of better diet quality [24], and better diet quality is suggested to be related to higher levels of EF, it is possible that the relation between breastfeeding and inhibitory control is (partially) mediated by habitual diet quality. Therefore, our goal was to investigate whether breastfeeding history (i.e., exclusive breastfeeding duration and breastfeeding cessation age) is predictive of inhibitory control at age 3 years, and if this association is (partially) mediated by habitual diet quality. As most previous studies investigated EF [18,19,25], for comparability purposes, this study investigated EF as a secondary outcome.

## 2. Materials and Methods

### 2.1. Participants

This study is part of the ongoing longitudinal *BINGO* study investigating early predictors of child development. Participants were healthy children living in the Netherlands, whose parents were recruited during pregnancy in 2014/2015. Prenatal exclusion criteria were excessive alcohol use, drug use, health problems or pregnancy complications, and insufficient knowledge of the Dutch language. At baseline, 88 pregnant women joined the *BINGO* study. Postnatal exclusion criteria were: complications during pregnancy (after initial contact), gestational age at birth <37 weeks, birth weight <2500 g, 5-min Apgar score <7, and congenital malformations. After postnatal exclusion, 77 mothers were followed up [26]. At the 36-month measurement round (2017/2018), 76 families were contacted, as one drop-out occurred during the previous measurement rounds. Six families did not participate due to time constraints, and one family dropped out due to personal reasons. Two families could not be contacted. There were no differences in parental demographics between participating and non-participating families. In total, 67 families participated. In 54 families (81%), both parents participated, and in 13 families (19%), only mothers participated. Four families were too busy to participate in a home visit and only filled in questionnaires. See Figure 1 for the flow chart of the *BINGO* study, leading to participants of the current paper. The *BINGO* study was approved by the Ethical Committee of the Faculty of Social Sciences of Radboud University [ECSW2014-1003-189 and amendment: ECSW-2018-034]. The current study was preregistered on the Open Science Framework: https://osf.io/5mgnf and amendment: https://osf.io/35tg6 accessed on 8 November 2019.

### 2.2. Data Collection Procedure

Information on exclusive breastfeeding duration (no additional solids or formula feeding next to breastfeeding) and breastfeeding cessation age (age when child stopped receiving breastfeeding) was collected via maternal questionnaires at two, six and 12 weeks, and at 12 and 36 months of age. At 36 months, a home visit took place in either the morning or afternoon. Children were fed before the home visit took place. The home visit consisted of: two inhibitory control tasks (Snack Delay and Flanker), saliva collection, prosocial task, mother-child interaction tasks, two inhibitory control tasks (Whisper and Bear Dragon), saliva collection, partner-child interaction tasks, prosocial task, and two inhibitory control tasks (Gift Wrap and Gift Delay). Only the inhibitory control tasks are part of the current study. Distractions, such as siblings, doorbells ringing or other factors were handled by the parent who was available (i.e., not performing a task with the child), or the student assistant who joined every home visit. Tasks were video recorded and afterwards rated by two trained observers. Additionally, mothers and partners independently filled in digital questionnaires about their child’s and their own inhibitory control and EF. Lastly, parents received three unannounced online 24 hours (24-h) recalls before the home visit to assess their child’s habitual nutritional intake. If parents had missed a 24-h recall, a recall was performed during the home visit, or was sent after the home visit.

### 2.3. Measures

#### 2.3.1. Breastfeeding History

At age 2, 6, and 12 weeks, mothers were asked if they exclusively breastfed the child. Water intake during exclusive breastfeeding was not asked. Note that in the Netherlands it is very uncommon to give exclusively breastfeeding infants water. At age 12 months, mothers reported their child’s feeding history: (1) are you breastfeeding? (yes or no; if no, breastfeeding cessation age was asked), (2) age of formula introduction, (3) age of solids introduction (i.e., fruits, vegetables, porridge). At age 36 months, mothers were again asked about the child’s breastfeeding history and to describe the child’s solid food intake. To accurately determine breastfeeding history, the breastfeeding data from the 36-months measurement follow-up was compared to that of the previous assessments. If data between the time points were conflicting (e.g., 3 months of breastfeeding reported at 12 weeks versus 2 months of breastfeeding reported at 36 months), the data with the shortest recall time were used, as retrospective data are more susceptible to recall bias [27]. Breastfeeding cessation age correlated significantly with exclusive breastfeeding duration (*r* = 0.64, *p* < 0.001). In line with earlier studies [18,19], the main analyses were run separately for both variables.

#### 2.3.2. Diet Quality

Three online 24-h food intake recalls, using *Compl-eat*™ [28], were used to measure children’s nutritional intake. Mothers were asked to report the dietary intake of their child from the previous day on two unannounced random weekdays and one unannounced random weekend day. Scores were given to each 24-h recall according to a diet quality score for preschool children [29]. The diet quality score is determined by the intake and amount of 10 dietary food groups (see Table 1).

A score between 0 and 1 was given to each food group. The ratio of the reported intake and the cut off level was calculated (scores were truncated at 1). For example, vegetable intake of 80 g per day was assigned a score of 0.8 (80/100 g/d). This was reverse-scored for intake of candy and snacks, and sugar-sweetened beverages. Scores for the 10 food groups were summed to create a diet quality score for each 24-h recall, and subsequently averaged to determine the overall diet quality score. A higher score corresponds with higher diet quality.

#### 2.3.3. Inhibitory Control Tasks

Behavioral tasks were chosen according to five categories of inhibitory control classified by Anderson and Reidy (2012) [30]: Delay of gratification (i.e., resist direct temptation to receive a bigger reward after the delay), Verbal inhibition (i.e., inhibit verbal responses), Go/No-go (i.e., perform certain behavior after being shown a stimulus and to inhibit that behavior after being shown a different stimulus), Motoric inhibition (i.e., learn response sets that conflict with an established behavior) and Impulse control (i.e., inhibit an instinctive response).

**Snack Delay** [31,32]. To measure delay of gratification, Snack Delay was used. Children were asked to put their hands on a placemat. Then, a self-chosen snack was placed at the top-center of the mat, and was covered with a transparent cup. Children were instructed to take the snack after the experimenter rang a bell. After a maximum of three practice trials, three consecutive trials were conducted with delays for ringing the bell of 20, 30 and 50 seconds, respectively. Children’s waiting behavior was coded every five seconds with a score ranging from 0 to 4 (0 = eats snack before the bell rings; 1 = touches/grasps snack before the bell rings; 2 = touches/grasps cup before the bell rings; 3 = waits for the bell to ring without hands on the placemat; 4 = waits for the bell to ring with hands on the placemat), and summed. Due to insufficient variation (no child ate the snack or touched the cup), this task was excluded from analyses.

**Flanker** [33]. Flanker was used to measure motoric inhibition. The Flanker task showed excellent test-retest reliability and excellent convergent validity in preschool aged children [34]. Children were asked to point in the same direction of where a centrally located target fish was swimming towards, ignoring the presence of interfering stimuli (flanking fish oriented in the same or opposite directions). After four practice trials, children were presented seven congruent trials and three incongruent trials. Accuracy in the incongruent trials was scored between 0 and 3 (0 = pointing in the wrong direction; 1 = first pointing correctly, then pointing in the wrong direction; 2 = first pointing wrongly, then pointing in the correct direction; 3 = pointing in the correct direction), and averaged. Out of 63 children tested, 49 passed the practice trial.

**Whisper** [31,32]. To measure verbal inhibition, Whisper was used. After two practice trials, children were asked to whisper the names of 12 presented animal pictures. Responses were coded 0 to 2 for every picture (0 = shout; 1 = normal or mixed tone; 2 = whisper), and averaged.

**Bear Dragon** [32,35]. To measure go/no-go, Bear Dragon was used. The experimenter introduced a “nice” bear puppet (using a soft, high-pitched voice) and a “naughty” dragon puppet (using a gruff, low-pitched voice). Children were told to obey the bear’s commands and ignore the dragon’s commands. After a maximum of three practice trials, 10 test trials followed. Child behavior was scored per dragon command, ranging from 0 to 2 (0 = obeying the dragon’s command; 1 = corrected movement to the dragon’s command; 2 = ignoring the dragon’s command), and averaged. Due to the low number of children (*n* = 31) that passed the practice trials, this task was excluded from the analyses. This low number is similar to a study by Kloo and Sodian (2017) [36], where more than 50% of the preschoolers failed the practice trials.

**Gift Wrap** [31,32]. To measure motoric inhibition, Gift Wrap was used. Children were asked to cover their eyes with their hands and not peek while their gift, in front of them, was being wrapped for one minute. Children’s waiting behavior was coded every five seconds with a score ranging from 0 to 3 (0 = watches wrapping/gift; 1 = peeks; 2 = looks away from wrapping/gift; 3 = closed eyes and/or hands in front of the eyes), and averaged. One child was unable to follow instructions for this task.

**Gift Delay** [32]. To measure impulse control, Gift Delay was used. Children were asked to not touch and unwrap the present, placed in front of the child, while the examiner left the room for 1 1/2 min. Latency (measured in seconds) until touching the present was used as a measure of impulse control. A higher score indicated better inhibitory control.

#### 2.3.4. Reliability of Coding

Recordings were observed by two observers independently. A codebook was made to set the rules for coding. The first five recordings were coded by both observers independently and immediately checked for agreement. Disagreements were discussed and adjusted in the codebook. Thereafter, the observers continued and only discussed recordings in case of insecurities. To determine inter-rater reliability, 30 out of 63 recordings were scored by both observers. Reliability was quantified by the Intraclass Correlation Coefficient (ICC) relying on absolute agreement. The ICC’s for the inhibitory control tasks were good, ranging from *r* = 0.84 to *r* = 0.96 (*p* < 0.001): 0.95 for the Flanker, 0.86 for the Whisper, 0.96 for the Bear Dragon, 0.88 for the Gift wrap, and 0.84 for the Gift delay.

#### 2.3.5. Parental Questionnaires on Inhibitory Control and Executive Functioning

**ECBQ** [37]. The Early Child Behavior Questionnaire (ECBQ) is a 107-item questionnaire of child temperament that was filled in by mothers only. The ECBQ contains a 6-item inhibition subscale, scored on a 7-point scale. A higher score indicates better inhibitory control. Because the Cronbach’s alpha was 0.59 for the inhibition subscale, this subscale was removed from further analyses.

**BRIEF-P** [38]. The Behavior Rating Inventory of Executive Function-Preschool Version (BRIEF-P) is a 63-item questionnaire of EF in preschool age, and contains a 16-item inhibition subscale, scored on a 3-point scale. As higher scores indicate worse EF and inhibitory control, and to align with our other inhibition and EF measures, the outcome of the BRIEF-P was reverse-coded. Consequently, higher scores on the BRIEF-P indicated better inhibitory control and EF. The Cronbach’s alphas were good for the inhibition subscale and the total EF score (mothers: α = 0.89 and α = 0.94, respectively, and partners: α = 0.84 and α = 0.96, respectively).

**REEF** [39]. The Ratings of Everyday Executive Functioning (REEF) is a 77-item questionnaire of EF in preschool age, using a 4-point scale. The REEF contains an inhibitory subscale, but usage of separate subscales appeared unreliable [39]. Therefore, only the total EF score was calculated. A higher score indicates better EF. The Cronbach’s alpha was 0.96 for mothers and 0.95 for partners.

#### 2.3.6. Confounding

The following confounding variables were taken into account: maternal educational level (ranging from 1, primary education, to 8, university education) [40,41], child gender (1 = boy, 2 = girl), and parental inhibitory control [40]. For this last confounder, parental inhibitory control was assessed with the Behavior Rating Inventory of Executive Function-Adult (BRIEF-A) [42]. The BRIEF-A is a 75-item self-report questionnaire of EF in adults, and contains an eight-item inhibition subscale, scored on a 3-point scale. We reverse-coded the BRIEF-A outcome for interpretation purposes, so that higher scores indicate better inhibitory control and EF. The Cronbach’s alphas for the inhibition subscale were insufficient for mothers (α = 0.57) and partners (α = 0.54). Therefore, the parental total score of the BRIEF-A was used as a confounder (Cronbach’s alpha: 0.96 for mothers, and 0.93 for partners). To preserve power, confounding variables were only included if they significantly correlated with the independent variables or the outcome variables [43].

### 2.4. Statistical Analyses

Descriptive analyses and normality analyses were computed. As not all variables were normally distributed, robust estimators were used in our main analyses. Furthermore, the data were inspected for outliers. Two variables contained one outlier each (breastfeeding cessation age, and the maternal BRIEF-P), and were subsequently winsorized [44]. Results were similar with and without winsorizing. Pearson and Spearman correlations were performed to correlate (non-)normally distributed variables. Sample size could not be adjusted due to the longitudinal nature of our study. According to Fritz and MacKinnon (2007) [45], with a power of 0.8, our study was able to detect medium to large mediation effects (β ≥ 0.39 and β ≥ 0.59). Robust estimators were used to account for small sample size [46].

### 2.5. Missing Data

The following 24-h recall data were missing: day one (*n* = 1), day two (*n* = 6) and day three (*n* = 12). Missing 24-h recall data were imputed by means of expected maximization to allow for calculation of the average diet quality score. The following behavioral data were missing: Whisper (*n* = 4), Flanker (*n* = 18, of which 15 were missing because children did not pass the practice trial), Gift Wrap (*n* = 5), and Gift Delay (*n* = 4). The following maternal questionnaire data were missing: REEF (*n* = 1). The following partner questionnaire data were missing: REEF (*n* = 16), BRIEF-P (*n* = 15) and BRIEF-A (*n* = 21) (of which 15 of each questionnaire were missing because partners did not join this study from the start). The LittleMCAR test from the BaylorEdPsych package indicated that data were missing completely at random (*X*^2^ = 272.547, *p* = 0.445). Missing data were accounted for by means of Full Information Maximum Likelihood (FIML) in the analyses.

### 2.6. Latent Variable and Composite Score Creation

**Latent variable**. Latent variables were computed when the following assumptions were met: Kaiser–Meyer–Olkin (KMO) >0.6 [47], Bartlett’s test of sphericity *p* < 0.05 [48], and linear independency *p* > 0.00001.

**Composite score**. Since we assume that the different inhibitory control tasks measure different forms of the same overarching construct, “lumping” was preferred over “splitting”. Therefore, and following our preregistration, if a latent variable could not be created due to violations of the assumptions, a composite score was made using z-scores.

### 2.7. Main Analyses

Structural Equation Modelling (SEM) was employed using the Lavaan package [49]. SEM has several advantages over standard regression models such as allowing for multiple independent variables to be added in one model and handling missing data [50]. The models were adjusted based on corresponding modification indices. For each modification, the covariance with the highest modification index was included in the previous model if it was theoretically logical. Models were adjusted handling the goodness of fit indices: Comparative Fit Index (CFI) > 0.95, Root Mean Square Error of Approximation (RMSEA) < 0.05, Standardized Root Mean Square Residual (SRMR) < 0.05 and a non-significant *X*^2^ (*p* > 0.05) [46]. Bias-corrected confidence intervals were obtained by use of bootstrapping, as it is recommended to check bootstrapped confidence intervals as well as significance when drawing conclusions [51].

To test whether breastfeeding history was predictive of inhibitory control, and if diet quality at age 3 mediated this association, four SEM models were run (see Figure 2 for a general path diagram illustrating our SEM models). For the first two models, we tested if exclusive breastfeeding duration (model 1) and breastfeeding cessation age (model 2) were predictive of observed inhibitory control determined by behavioral tests. In model 3 and 4, reported inhibitory control (BRIEF-P inhibition subscale) was used as the outcome measure.

### 2.8. Comparability Analyses

For comparability purposes, we investigated whether exclusive breastfeeding duration (model 5) and breastfeeding cessation age (model 6) were related to EF at age 3, determined by a latent score of the BRIEF-P and the REEF. Because models 5 and 6 could not be fitted in the SEM model, an additional four models (models 7 to 10) were tested. We tested if exclusive breastfeeding duration (model 7) and breastfeeding cessation age (model 8) were predictive of reported EF determined by the BRIEF-P composite score, and whether exclusive breastfeeding duration (model 9) and breastfeeding cessation age (model 10) were predictive of reported EF determined by the REEF composite score. In model 5 to 10, the mediating role of diet quality at age 3 was also assessed.

### 2.9. Exploratory Analyses

Exploratorily, we tested if exclusive breastfeeding duration and breastfeeding cessation age were predictive of individual inhibitory control tasks, and whether diet quality at age 3 mediated these associations. Furthermore, we investigated the mediating role of three different food groups (vegetable, fruit, and snacks and candy) in the association between exclusive breastfeeding duration and breastfeeding cessation age, and inhibitory control and EF. Fish was not investigated because of low variation of fish intake.

## 3. Results

### 3.1. Preliminary Analyses

#### 3.1.1. Descriptives

Table 2 presents descriptive statistics of the study population. The study population is mostly highly educated. Scores on questionnaires differed significantly between mothers and partners for the BRIEF-P and the REEF. Average diet quality score was a 3.9, which is similar to the diet quality score from a previous study in Dutch children between 12–28 months (i.e., 4.1; Voortman et al., 2015) [29].

Table 3 shows correlations between the study variables. Longer duration of exclusive breastfeeding is significantly correlated with cessation of breastfeeding (*r* = 0.638, *p* < 0.001), and with better diet quality at age 3 (*r* = 0.321, *p* = 0.009). Scores of the inhibitory control subscale from the BRIEF-P correlated significantly between mother and partner (*r* = 0.415, *p* = 0.002). There were no significant correlations between behavioral tasks. Appendix A shows the correlations between mother and partner reports.

#### 3.1.2. Latent Variable and Composite Score Creation

**For the main analyses**. Assumption testing for latent variable creation was performed for the behavioral tasks: Flanker, Whisper, Gift Wrap, and Gift Delay. KMO values ranged between 0.41 and 0.55. Bartlett’s test did not reach significance, and determination testing yielded *p* = 0.934. Because KMO values and Bartlett’s test did not meet the cut off values, a latent variable for the behavioral tasks was not created. Instead, a composite score was made by averaging z-scores of the inhibitory control tasks. An average score was calculated only if maximally one test score was missing. In total, four children had more than one test score missing.

Next, we performed assumption testing for the BRIEF-P inhibition subscale filled in by mother and partner. KMO values were 0.5 for maternal and partner reports. Bartlett’s test reached significance, and determinant testing yielded results *p* > 0.0001. Because KMO values were lower than 0.6, a latent variable was not created. Instead, a composite score was made for the BRIEF-P inhibitory control score by averaging parental scores.

**For the comparability analyses**. To define reported EF for the comparability analyses, assumption testing for latent variable creation was performed from the four EF questionnaires (BRIEF-P and the REEF, filled in by mother and partner). KMO values were: BRIEF-P of mother: 0.59, and partner: 0.62, REEF of mother: 0.62, and partner: 0.61. Bartlett’s test reached significance (*p* < 0.001), and the data was linearly independent (*p* = 0.488). The KMO value for the maternal BRIEF-P was 0.59, indicating borderline acceptable sampling adequacy. However, because the KMO value for the other three questionnaires was acceptable, and creating a latent variable was preferred over a composite score, we created a latent variable from the four parental EF questionnaires. Nonetheless, the models with the latent variable for EF could not be fitted (models 5 and 6). Therefore, assumption testing for latent variable creation was performed to combine mother and partner scores of the BRIEF-P, and to combine mother and partner scores of the REEF. The KMO value was 0.5 for all questionnaires, Bartlett’s test reached significance for all analyses, and data were linearly independent (*p* > 0.0001). Because the KMO values were lower than 0.6, latent variables were not created. Instead, composite scores of mother and partner questionnaires were made for the BRIEF-P and the REEF, which were used as outcome measures (models 7 to 10). Goodness of fit measures showed an adequate fit for models utilizing the BRIEF-P composite score as outcome variable (models 7 and 8), but not for models utilizing the REEF composite score as outcome variable (models 9 and 10).

#### 3.1.3. Confounders

Maternal educational level correlated significantly with breastfeeding cessation age (*r* = 0.290, *p* = 0.019), and was added as a confounder in the models where breastfeeding cessation age was used as predictor.

The BRIEF-A positively correlated with the BRIEF-P inhibition subscale (*r* = 0.351, *p* = 0.004) and the BRIEF-P total questionnaire (*r* = 0.437, *p* < 0.001), and negatively correlated with observed inhibitory control (*r* = −0.250, *p* = 0.048). The BRIEF-A was added to the model as a confounder when the BRIEF-P inhibition subscale, BRIEF-P total questionnaire, and observed inhibitory control were used as outcome variable. An overview of all correlations with potential confounding variables, including correlations with exploratory outcome measures, is shown in the Appendix A.

### 3.2. Main Analyses

Goodness of fit measures showed an adequate fit of final models 1 through 4. Parameter estimates and bootstrapped confidence intervals per model are shown in Table 4. Exclusive breastfeeding duration (model 1) and breastfeeding cessation age (model 2) did not predict observed inhibitory control. However, longer exclusive breastfeeding duration was associated with better diet quality at age three (*β* = 0.173, 95% CI [0.035, 0.310]). Next, exclusive breastfeeding duration (model 3) and breastfeeding cessation age (model 4) did not predict reported inhibitory control. Additionally, no significant mediation effect of diet quality score was found for models 1 to 4. Furthermore, better parental EF was associated with low levels of observed child inhibitory control at age 3 (*β* = −0.009, 95% CI [−0.017, −0.001]). Contrarily, better parental EF was associated with high levels of child inhibitory control reported by both parents at age 3 (*β* = 0.091, 95% CI [0.034, 0.148]).

### 3.3. Comparability Analyses

Results showed that exclusive breastfeeding duration (model 7) and breastfeeding cessation age (model 8) were not predictive of EF as reported by the BRIEF-P. Diet quality had no mediating effect in both models. Parameter estimates and their respective confidence intervals are shown in the Appendix A.

### 3.4. Exploratory Analyses

Breastfeeding cessation age was significantly associated with the Flanker task (*β* = 0.026, 95% CI [0.003, 0.048]). No other significant relations were found between breastfeeding history and the behavioral tasks. Parameter estimates and bootstrapped confidence intervals are shown in Appendix A. With respect to the food groups (vegetable, fruit, and snacks and candy), because all exploratory models could not be fitted using the latent variable for EF, the BRIEF-P composite score and the REEF composite score were used as outcome variables in separate models. No significant mediation effects of food groups were found (see Appendix A for the parameter estimates).

## 4. Discussion

This study did not provide evidence that breastfeeding history (i.e., exclusive breastfeeding duration and breastfeeding cessation age) is associated with inhibitory control and EF in 3-year-old children. Furthermore, diet quality at age 3 was not associated with inhibitory control and EF in 3-year-old children, and as such did not mediate a link between these child outcomes and breastfeeding history. Longer exclusive breastfeeding duration, but not breastfeeding cessation age, was predictive of higher diet quality at age 3.

Contrary to our hypotheses, breastfeeding history did not predict inhibitory control or EF in 3-year-olds. These results are in line with the results from Belfort et al. (2016) [19], and Lopez et al. (2021) [20], who also found no evidence for an association between breastfeeding and executive functioning. Contrarily, Julvez et al. (2007) [18] found that breastfeeding was associated with high levels of observed EF in a sample of 500 4-year-old children. The observational tasks used in Julvez et al.’s (2007) [18] study focused on other aspects of EF, such as working memory and attention, rather than inhibitory control specifically. Although we did not find an association between breastfeeding history and inhibitory control or the more general measures of EF, it remains possible that breastfeeding history is related to other specific aspects of EF.

Another explanation for our null-results regarding breastfeeding history might be that small amounts of breastfeeding are already beneficial for developing inhibitory control. Of the 500 mothers in Julvez et al.’s (2007) study, 196 breastfed their infants for less than 12 weeks. As our sample contained many mothers breastfeeding for a long period, and few mothers breastfeeding for less than 12 weeks (*n* = 7) or not breastfeeding (*n* = 5), future studies investigating inhibitory control in a sample with more variation in breastfeeding duration may help shed light on these differences in results.

No evidence was found that habitual diet quality predicted inhibitory control and EF, or mediated the relation between breastfeeding history and these outcomes. Research on habitual diet has been performed more often in older children (6–18 year) in relation to ADHD and EF [10,52]. Moreover, of the two studies that assessed nutritional intake in relation to inhibitory control in preschool-aged children, one found that higher intake of sugars was associated with worse inhibitory control [22], while the other found no association between nutritional intake and inhibitory control [21]. Both these studies assessed spontaneous food choices in an experimental setting, while our study is the first to study habitual diet and inhibitory control in preschool age. This makes comparisons difficult and stresses the need for further studies in preschool-aged children, preferably replication studies using the same measurement methods to assess dietary intake, inhibitory control, and EF.

It might also be possible that the effects of breastfeeding and diet quality are different between children due to moderating effects by individual differences. One such factor that could moderate the association between diet and child outcomes, that has recently been receiving attention, is the gut microbiota. The gut microbiota are a collection of bacteria, archaea and eukarya colonizing the gastrointestinal tract of the human [53]. Recent studies showed that individuals with ADHD—who have lower levels of inhibitory control—have different bacteria present in their gut compared to individuals without ADHD [54,55]. Since gut microbiota are in constant interplay with diet, it is possible that the gut microbiota moderate the effects of diet on the brain via the microbiota-gut-brain axis [11,12]. As gut microbiota composition differs between 3-year-old children [56], it may affect the way nutrition impacts brain development in individual children [57].

Furthermore, our results indicated that longer exclusive breastfeeding duration is predictive of better diet quality at preschool age. This is consistent with Ventura′s (2017) review [24] which included studies investigating food groups and showed that longer exclusive and total breastfeeding duration is associated with higher intake of vegetables and fruits in young children, contributing to better diet quality. Note, however, that we did not find an association between breastfeeding cessation age and diet quality. Potentially, exclusive breastfeeding duration is a better indicator of maternal investment in the quality of offspring nutrition than breastfeeding cessation age. Because high diet quality is also associated with different child outcomes, including improved physical, school and emotional functioning, and psychosocial quality of life [58], we recommend future studies to consider the role of diet quality when studying relations between breastfeeding and child outcomes.

Lastly, we found that the behavioral tasks in our study did not correlate. Previous studies also found no consistent correlations between multiple inhibitory control tasks [31,59]. Following the categorization of Anderson and Reidy (2012), who divided inhibitory control into motoric inhibition, verbal inhibition, impulse control, go/no-go, and delay of gratification, we deliberately choose our tasks to measure each of these components. Though this might explain the absence of correlations to some part, if all components tap onto the same overarching construct, some correlations would have been expected. The question arises whether the four tasks that we used accurately measure inhibitory control, and how other factors play a role, such as age, attention and cognitive functioning [60]. Therefore, we encourage future research to reinvestigate tasks to be able to accurately measure inhibitory control in preschool aged children.

Our study has several strengths including the longitudinal nature, covering the period from pregnancy and birth until 3 years and allowing for frequent maternal reports on breastfeeding status, the assessment of inhibitory control using multiple behavioral tasks and reports by mother and partner, and the assessment of habitual diet with three 24-h recalls (i.e., considered the least biased self-report instrument; Thompson and Subar, 2017 [61]). The current study also has limitations. The generalizability of our findings is limited by our mostly highly educated sample and low variation in breastfeeding duration. Furthermore, the relatively small sample size reduced our statistical power. We preserved power by using composite scores and performing exploratory analyses for individual scores, but replications in larger and more varied study populations are highly recommended.

## 5. Conclusions

To conclude, no evidence was found for a relation between breastfeeding history and inhibitory control or EF, and no evidence was found for a mediating effect of diet quality. Nonetheless, we found that longer exclusive breastfeeding duration predicted better diet quality at age 3, which is similar to previous literature. The fact that we found no evidence for the expected association between breastfeeding history, habitual diet, and inhibitory control/EF, is not equivalent to confirming the null hypothesis, especially considering our study limitations. Given our results and the inconclusive nature of previous literature, as well as the role of low inhibitory control in psychopathology and child behavior problems, we recommend future studies to continue investigating nutrition and inhibitory control in the toddler to preschool ages.

## Figures and Tables

**Figure 1 nutrients-13-02752-f001:**
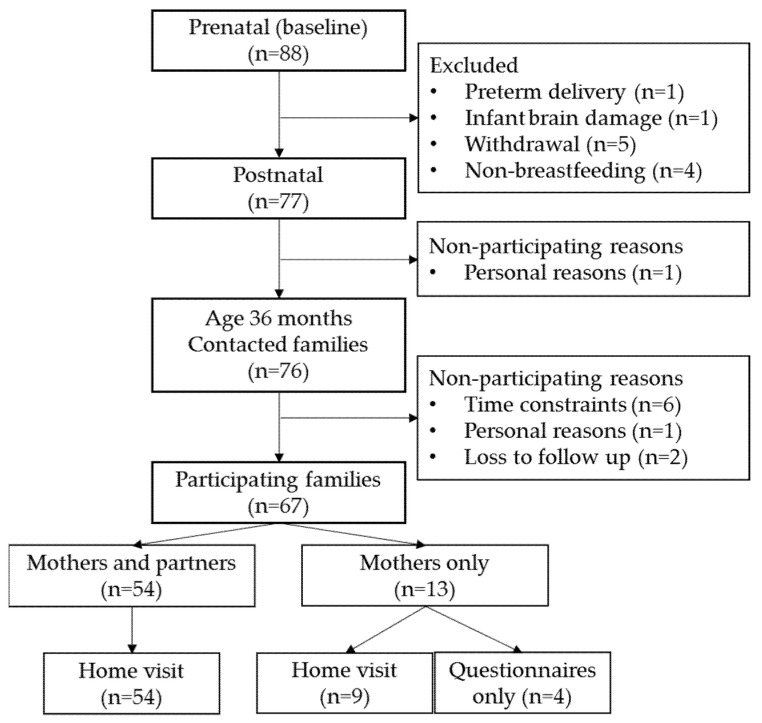
Flow chart of participating families.

**Figure 2 nutrients-13-02752-f002:**
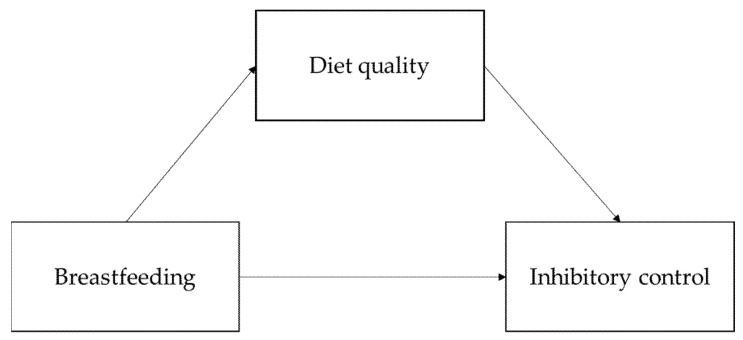
General path diagram of the SEM models.

**Table 1 nutrients-13-02752-t001:** Food groups and cut off levels.

Food Group	Cut-off Level
Vegetables	≥100 g/d
Fruit	≥150 g/d
Bread and cereals	≥70 g/d
Rice, pasta, potatoes, and legumes	≥70 g/d
Dairy	≥350 g/d
Meat, eggs and meat substitutes	≥35 g/d
Fish	≥15 g/d
Oils and fats	≥25 g/d
Candy and snacks	≤20 g/d
Sugar-sweetened beverages	≤100 g/d

Cut off levels are determined by Voortman et al. (2015) [29]; g/d: grams per day.

**Table 2 nutrients-13-02752-t002:** Characteristics of the study population.

	Mean ± SD	Range	*n*	
**Maternal characteristics**				
Age (years)	34.4 ± 3.7	28–44	67	
Educational level			65	
Low	0%			
Middle	13.8%			
High	86.2%			
**Partner characteristics**				
Gender			51	
Male	96.1%			
Female	3.9%			
Age (years)	35.8 ± 4.1	28–50	51	
Educational level ^a^			48	
Low	4.2%			
Middle	16.6%			
High	79.2%			
**Child characteristics**				
Sex			67	
Boy	47.8%			
Girl	52.2%			
Child birthweight (grams)	3531.8 ± 420.0	2570–4445	63	
Child gestational age (weeks)	39.8 ± 1.6	35.6–42.1	67	
Child age (months)	37.7 ± 1.2	36–47	67	
**Study variables**				
Breastfeeding				
Exclusive breastfeeding duration (months)	3.5 ± 1.9		67	
Breastfeeding cessation age (months)	9.5 ± 8.0		67	
Diet quality	3.9 ± 1.1			
Behavioral tests				
Flanker	1.2 ± 0.7		49	
Whisper	1.8 ± 0.3		63	
Gift Wrap	2.1 ± 0.9		62	
Gift Delay (seconds)	77.9 ± 27.5		63	
Questionnaires	Mother	Partner	n (mother)	n (partner)
BRIEF-P inhibitory control scale	23.1 ± 5.8	22.8 ± 4.6 ***	67	52
BRIEF-P	94.6 ± 15.6	92.4 ± 18.0 ***	67	52
REEF	147.8 ± 33.1	146.6 ± 28.8 ***	66	51

SD: Standard deviation; ^a^: assessed during pregnancy. *** *p* < 0.001.

**Table 3 nutrients-13-02752-t003:** Correlations between questionnaires, breastfeeding history, and diet score.

	Exclusive BF Duration	Age of BF Cessation	Diet Quality Score	Flanker	Whisper	Gift Wrap	Gift Delay	BRIEF-P inh-M	BRIEF-P inh-P
Exclusive BF duration	-								
Age of BF cessation	0.638 **	-							
Diet quality score	0.321 **	0.117	-						
Flanker	0.161	0.341 *	−0.105	-					
Whisper	−0.117	−0.107	0.063	−0.069	-				
Gift wrap	−0.137	0.029	0.177	0.134	−0.117	-			
Gift Delay	0.123	0.212	−0.036	0.137	0.135	0.211	-		
BRIEF-P inh-M	0.115	0.139	0.056	−0.065	0.250 *	0.052	0.152	-	
BRIEF-P inh-P	0.219	0.078	0.196	0.033	0.051	−0.081	−0.012	0.415 **	-

Correlations are denoted as *r*. BF: Breastfeeding; BRIEF-P-inh-M: Score of the inhibitory control scale of the BRIEF-P filled in by mother; BRIEF-P-inh-P: Score of the inhibitory control scale of the BRIEF-P filled in by partner.* *p* < 0.05. ** *p* < 0.01.

**Table 4 nutrients-13-02752-t004:** Parameter estimates and bootstrapped confidence intervals for Model 1, 2, 3, and 4.

	B	SE	Lower CI	Upper CI
**Regression Paths**	Model 1: Exclusive breastfeeding duration → Diet quality score → Observed inhibitory control			
Observed inhibitory control (composite)				
Exclusive breastfeeding duration	−0.016	0.035	−0.085	0.052
Diet quality score	0.065	0.066	−0.063	0.194
Parental executive functioning	−0.009	0.004	−0.017	0.000
Diet quality score				
Exclusive breastfeeding duration	0.173 *	0.070	0.035	0.311
**Mediation effect**	0.011	0.012	−0.013	0.035
**Total effect**	−0.013	0.033	−0.078	0.051
**Regression Paths**	Model 2: Breastfeeding cessation age → Diet quality score → Observed inhibitory control			
Observed inhibitory control (composite)				
Breastfeeding cessation age	0.008	0.011	−0.013	0.029
Diet quality score	0.024	0.057	−0.088	0.136
Parental executive functioning	−0.009 *	0.004	−0.017	−0.001
Maternal education	0.089	0.048	−0.004	0.182
Diet quality score				
Breastfeeding cessation age	0.024	0.016	−0.008	0.055
**Mediation effect**	0.001	0.001	−0.002	0.003
**Total effect**	0.088	0.047	−0.003	0.180
**Regression Paths**	Model 3: Exclusive breastfeeding duration → Diet quality score → Reported inhibitory control			
Reported inhibitory control				
Exclusive breastfeeding duration	0.296	0.339	−0.368	0.961
Diet quality score	−0.167	0.469	−1.086	0.751
Parental executive functioning	0.086 **	0.030	0.028	0.144
Diet quality score				
Exclusive breastfeeding duration	0.172 *	0.070	0.035	0.310
**Mediation effect**	−0.029	0.0832	−0.190	0.132
**Total effect**	0.354	0.309	−0.252	0.959
**Regression Paths**	Model 4: Breastfeeding cessation age → Diet quality score → Reported inhibitory control			
Reported inhibitory control				
Breastfeeding cessation age	0.114	0.070	−0.024	0.252
Diet quality score	−0.125	0.443	−0.992	0.743
Parental executive functioning	0.091 **	0.029	0.034	0.148
Maternal educational level	−0.052	0.389	−0.815	0.711
Diet quality score				
Breastfeeding cessation age	0.024	0.016	−0.008	0.055
**Mediation**	−0.003	0.011	−0.025	0.019
**Total model**	0.150	0.369	−0.573	0.873

MLR estimator used to calculate parameter estimates, bootstrapping used to calculate bias-corrected confidence intervals. Model 1: *X*^2^ (3) = 0.919, *p* = 0.338; CFI = 1.000, RMSEA = 0.000, SRMR = 0.029, *n* = 66. Model 2: *X*^2^ (4) = 1.804, *p* = 0.614; CFI = 1.000, RMSEA = 0.000, SRMR = 0.045, *n* = 66. Model 3: *X*^2^ (3) = 0.996, *p* = 0.307; CFI = 0.996, RMSEA = 0.026, SRMR = 0.031, *n* = 67. Model 4: *X*^2^ (4) = 1.779, *p* = 0.619; CFI = 1.000, RMSEA = 0.000, SRMR = 0.044, *n* = 67. * *p* < 0.05. ** *p* < 0.01.

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
