# Peer review of "Do Breastfeeding History and Diet Quality Predict Inhibitory Control at Preschool Age?"

_nutrients, 2021, doi:10.3390/nu13082752_

Round 1
Reviewer 1 Report
An overall well written manuscript but with major methodological issues that can't be changed (i.e. no variation in breastfeeding duration, small sample size).
Abstract and Introduction
- In the abstract, what is meant by preregistered? I would suggest removing this, or changing to another word.
- In the background, I find that some important information is missing. For example, when discussing previous research investigating EF and breastfeeding, it would be important to include what type of epidemiologic study was conducted, and the study population. Please revise the background with this in mind.
- I also think it would be critical to understand the biological pathway. Please include any potential biological explanations into why your investigated relationship could exist.
- Please also include the definition of inhibitory control.
Methods
- Although you mention where we can find the inclusion criteria, this is critical for a readers understanding about the methods. Please include in your paper instead of referring elsewhere.
- The participants section is very difficult to follow. I would suggest creating a figure which displays this information. You could consider a CONSORT figure.
- Please include the years of data collection
- Please include if water was allowed for exclusive breastfeeding
- The way in which breastfeeding duration was calculated needs further explanation. Why was the shortest recall used. Is there evidence that this method provides the most accurate duration.
- There was a lot of measures used, which is great that you have this information; however, I am unsure if they are high quality. Can you include evidence showing the quality and reliability of each of these measures.
- The confounding list is very limited, and a major weakness of the study. Please include a longer list of confounding factors. Otherwise, the results will not be reliable.
- I do not understand the difference between breastfeeding duration and cessation age. These seem the same to me.
- I would consider using the latent class analysis, to then create a new variable and run a linear regression or other model with more robust statistical models.
Results
- Please include in the limitations section your small sample size.
Discussion
- Due to the limited sample size, and the lack of additional statistical analyses, please tone down the study’s conclusions
- The lack of variation between breastfeeding duration is another major limitation.
Reviewer 2 Report
Thank you for the opportunity to review the manuscript “Do breastfeeding history and diet quality predict inhibitory 2 control at preschool age?” (nutrients-1299318) that aims to investigate whether breastfeeding history (i.e. exclusive breastfeeding duration and breastfeeding cessation age) is predictive of inhibitory control at age 3 years, and if this association is (partially) mediated by habitual diet quality.
This is a relevant and very well written manuscript and I strongly recommend publication. Some minor revisions could improve the paper:
Line 78
“As most previous studies investigated EF [16,17,23], for generalization purposes, this study investigated EF as a secondary outcome.” I believe you mean comparability and not generalilzation
Line 88
Please clarify why you only contacted 76 families. As far as I understand from reading the Hechler, Beijers, Riksen-Walraven, & de Weerth (2018), there were more than 76 eligible mothers.
Line 103
I believe you have repeated some of the measures?
“The home visit consisted of: two inhibitory control tasks, saliva collection, prosocial task, mother-child interaction tasks, two inhibitory control tasks, saliva collection, partner-child interaction tasks, prosocial task, and two inhibitory control tasks”
Line 130
Did parents complete the questionnaires separately? Or only one answer per child? Please clarify
Line 147 and lines 153/176
Since you did not include the snack delay and the Bear dragon in the analysis, I do not think it is necessary to describe those tasks. You can indicate in line 147 that the delay of gratification and the Go/No/Go tasks were performed but not included and explain reasons
Line 171
I do nor understand what you mean by “Out of … children tested”, please clarify
Line 200-201
Please clarify if the information on the ICC is only regarding the tasks included.
“The ICC’s for the inhibitory control tasks were good, ranging from r=0.84 to r=0.96 (p<0.001).”
You included 4 tasks in the analysis, so I recommend discriminating the ICC by task.
Line 204
Since you did not used the ECQ in the analysis I recommend that you do not describe it.
Line 285
Please clarify what you mean by “(BRIEF-P inhibition subscale, mother and partner combined)”. It looks like you computed a mean score out of the mother and the partner questionnaires? If so, please provide that information in lines 209 to 216.
Line 287
I really would not call it generalization analyses. This is other equally legitime aim of your paper, that will not allow generalization, but will allow comparability: “investigate whether breastfeeding history (i.e. exclusive breastfeeding duration and breastfeeding cessation age) is predictive of EF at age 3 years”.
Still we usually do not classify our analysis by what they will allow us to do, but by which research objective/question they will allow to respond. Additionally, I really think this is extremely important and should not be considered secondary analysis (nor included in supplementary tables). I really think you should consider them as main analysis. In the discussion section you give the same importance to inhibitory control and to EF. So for coherence I would either consider the EF analysis as primary analysis (preferable), or maintain the EF analysis as secondary and focus the discussion on inhibitory control
Lines 262, 271, 287, 298, you have (1) Latent variable and composite score creation , (2) Main analyses, (3) Generalization analyses and (4) Exploratory analyses, respectively, but this is not the order in which you report the results. I recommend using the same order in the methods and in the results for coherence.
Line 383
If the age of cessation of breastfeeding (independent variable) is not associated (information provided in table 3) with the quality of diet (mediator), there is no mediation model to test in the first place.
Lines 384-387
It would be interesting to discuss the results reported in lines 384-387 in the discussion section
Line 459-469
Discussion on the association between breastfeeding and
Line 474
You did not used all of those tasks. Please focus on those that entered in the analysis.
Please provide information regarding the training of the researchers on scoring these tasks and detail the care that the researchers had to comply with the administration and scoring instructions. Clarify that the proper conditions in the home visits guarantee (the child was fed, no distractors and so on). If you report that all was guarantee you strengthen your argument on the reinvestigation of the tasks.
Reviewer 3 Report
Do breastfeeding history and diet quality predict inhibitory control at preschool age?
By Willemsen et al.
Willemsen and colleagues investigated whether breastfeeding history and habitual diet quality mediate inhibitory control behaviors in 3-year-old children. They found no apparent links between them. The investigations are well-executed and the findings are important to the field. One potential general weakness of the study could be that the authors focused only on inhibitory control as outcomes of executive functioning; however, the authors are well aware of this potential pitfall and discussed it clearly in the Discussion. I think the paper can be published without significant changes.
